# Characterization of Vegetative *Bacillus cereus* and *Bacillus subtilis* Strains Isolated from Processed Cheese Products in an Italian Dairy Plant

**DOI:** 10.3390/foods10112876

**Published:** 2021-11-21

**Authors:** Angela Maria Catania, Tiziana Civera, Pierluigi Aldo Di Ciccio, Maria Ausilia Grassi, Patrizia Morra, Alessandra Dalmasso

**Affiliations:** Department of Veterinary Sciences, University of Turin, Largo Braccini 2, Grugliasco, 10095 Turin, Italy; angelamaria.catania@unito.it (A.M.C.); pierluigialdo.diciccio@unito.it (P.A.D.C.); auxilia.grassi@unito.it (M.A.G.); patrizia.morra@unito.it (P.M.); alessandra.dalmasso@unito.it (A.D.)

**Keywords:** spoilage, *Bacillus cereus*, *Bacillus subtilis*, MLST, toxins, biofilm

## Abstract

Processed cheese is a commercial product characterized by high microbiological stability and extended shelf life obtained through the application of severe heat treatment. However, spore-forming bacteria can survive through thermal processes. Among them, microorganisms belonging to *Bacillus* genus have been reported. In this study, we examined the microbiological population of the first hours’ production of processed cheeses in an Italian dairy plant during two seasons, between June and October 2020. Matrix-assisted laser desorption ionization time-of-flight mass spectrometry (MALDI-TOF MS) was used to identify bacteria colonies, allowing the isolation of *Bacillus cereus* and *Bacillus*
*subtilis* strains. These results were further confirmed by amplification and sequencing of 16 rRNA bacterial region. A multi-locus sequence type (MLST) analysis was performed to assess the genetic similarity among a selection of isolates. The fourteen *B. cereus* strains showed two sequence types: ST-32 was observed in only one strain and the ST-371 in the remaining thirteen isolates. On the contrary, all twenty-one *B. subtlis* strains, included in the study, showed a new allelic profile for the *pycA* gene, resulting in a new sequence type: ST-249. For *B. cereus* strains, analysis of toxin genes was performed. All isolates were positive for *nheABC*, *entFM*, and *cytK*, while *hblABCD*, *bceT*, and *ces* were not detected. Moreover, the biofilm-forming ability of *B. cereus* and *B. subtilis* strains was assessed, and all selected isolates proved to be biofilm formers (most of them were stronger producers). Considering the genetical similarity between isolates, jointly with the capacity to produce biofilm, the presence of a recurring *Bacillus* population could be hypothesized.

## 1. Introduction

Processed cheeses are produced by melting into a homogeneous molten blend, one or more cheeses with other ingredients (cream, butter, whey) and usually emulsifying salts [1,2]. The mixture is then blended, heated, and portioned, according to the commercial use. The temperature and the time of treatments varies depending on the formulation, the desired product texture, and shelf-life characteristics. Processing may be carried out in batch cookers in which temperature–time treatment varies (e.g., 70–95 °C for 4–15 min), by obtaining pasteurized processed cheese or in continuous cookers in which the blend is typically heated with direct steam injection to ∼140 °C for ∼10 s, or by manufacturing sterilized high-moisture processed cheese [1].

The thermal processes allow the inactivation of most vegetative forms, but bacterial spores can survive [3]. Spore-forming bacteria are ubiquitous, founded mainly in the soil, but they are also able to colonize the gastrointestinal tract of insects and animals [4]. They belong to the Firmicutes phylum and, among them, Bacilli and Clostridia are the most relevant in the dairy industry [5]. Due to their ubiquity in nature, it is a real challenge to determine the true source of contamination in a dairy farm, which could be related to inadequate handling of raw materials [4], to cross contamination in the dairy plant [6], or for production of spores during food processing [7].

Moreover, it has been reported that, at certain conditions, the thermal treatment can stimulate the germination [8]. Once the germination process is triggered, the spores can return to a vegetative form and restart exponential cell division [9]. If spore germination occurs in a final product, followed by growth of vegetative cells, the consequence could be food spoilage or a safety issue if contamination due to pathogenic bacteria occurs [10,11].

The presence of spore-forming bacteria can negatively affect product quality [4]. While Clostridia are often associated to the late blowing defect in hard cheeses [12], Bacilli, in particular members of the *B. cereus* group, can produce several heat-stable extracellular enzymes, such as proteases and lipases [13]. The production of protease was also reported for *B. subtilis* [14]. Proteases can lead to bitter flavor and milk gelation [15], while lipases are often associated with a rancid flavor [16].

Among aerobic spore formers bacteria, *B. cereus* is also known for its potential to produce toxins related to two types of food-borne illness, i.e., the emetic and diarrheal syndrome [17]. It has been reported, although rarely, that other *Bacillus* species, such as *B. licheniformis* [18], *B. pumilus* [19], and *B. amyloliquefaciens* [20], have occasionally been reported as causative agents in food poisoning.

Moreover, *Bacillus* spp. can adhere to materials used in dairy process, forming biofilms [21], which are much more resistant to cleaning and disinfection processes than planktonic cells. This may represent a problem in processing equipment used in the dairy industry because may cause a continuous source of contamination and may affect the quality and safety of foods [22].

Processed cheeses are widely traded, thanks to their versatility in cooking preparations. Despite their broad use, few works have been focused on this product.

The aim of the present research was to explore and characterize the residual vegetative microbial population of processed cheeses produced in an Italian dairy plant.

## 2. Materials and Methods

### 2.1. Experimental Design and Sample Collection

A dairy cheese processing plant located in northern Italy served as a pilot site for this research. Hard cheeses, mainly cheddar, are used in addition to milk, butter, emulsifying salts, and other ingredients to produce processed cheeses, as shown in Figure 1.

Thirty-two batches of processed cheeses were analyzed during eight weeks, from June to October 2020. Sixteen batches were collected in summer and the other sixteen in autumn. For each batch, we examined the first 300 g produced within the first hours of processing, since the characteristics of the production line direct towards a possible greater contamination of the product during the initial phases of production.

### 2.2. Isolation and Enumeration of Bacterial Colonies

Twenty-five grams of cheese samples were suspended with 225 mL of a sterile physiological saline solution and peptone (Oxoid, Basingstoke, UK) using the Stomacher^®^ 400 Circulator (Seward Limited, Worthing, UK) for 1 min at a medium speed. The homogenate was serially diluted, and the count of vegetative cells was performed.

For enumeration and colonies identification of aerobic bacteria each dilution was spread-plated in duplicate onto a plate count agar (Oxoid, Basingstoke, UK), incubated at 30 °C and 37 °C for 24 h/48 h.

For presumptive anaerobic microorganisms, iron sulfite agar (Oxoid, Basingstoke, UK) was used, and the plates, in duplicate, were incubated under anaerobic conditions at 37 °C and 50 °C for 24 h/48 h.

### 2.3. DNA Extraction

For DNA extraction, isolated conies were inoculated in brain heart infusion (BHI) broth (Oxoid, Basingstoke, UK) and incubated overnight at 30 °C. DNA extraction was performed using the DNeasy Blood and Tissue Kit (Qiagen Ltd., Manchester, UK), according to the manufacturer’s instructions. The lysis step was carried out overnight at 37 °C with enzymatic lysis buffer (20 mM Tris·Cl, pH 8.0, 2 mM sodium EDTA, 1.2% Triton^®^ X-100) and a solution of freshly prepared lysozyme (20 mg/mL). DNA samples were eluted in 50 μL of 10 mM Tris·Cl pH 8.0 and stored at −20 °C until used. DNA yield and purity were determined using Nano drop 2000 spectrophotometer (Thermo Fisher Scientific, Waltham, MA, USA).

### 2.4. Isolated Colonies Identification by Matrix Assisted Laser Desorption Ionization-Time of Flight Mass Spectrometry (MALDI-TOF MS) and 16S rRNA Sequencing

The identification of single colonies grown on plate count agar was performed using matrix-assisted laser desorption–ionization time-of-flight mass spectrometry (MALDI-TOF MS) analysis in a Microflex LT MALDI Biotyper mass spectrometer (Bruker Daltonics, Hamburg, Germany).

For each plate, at least five colonies were selected according the morphotypes. Briefly, colonies were directly spotted on the MALDI plate and treated with 1 μL of formic acid and then overlaid with 1 μL of saturated α-cyano-4-hydroxycinnamic acid (HCCA) matrix solution which was then allowed to dry at room temperature. Each colony was tested in duplicate. The acquired mass spectra were imported into the MALDI Biotyper software (version 3.1) (Bruker Daltonics, Hamburg, Germany) and compared to a collection of different spectra stored in a database. The identification of an organism is expressed as a score based on pattern matching between samples and reference spectra. The degree of matching is expressed as a logarithmic score with values ≥ 2.00 to indicate high confidence species identification. Scores of 1.70–1.99 and <1.70 indicate, respectively, low confidence and no organism identification [23]. Only isolates with the highest score value of mass spectra were kept frozen at −20 °C, and considered for further analysis.

As a complementary approach to MALDI-TOF MS, sequencing of the 16S rRNA gene was performed to confirm strains identification. Bacterial DNA was amplified by PCR using 16S rRNA gene targeting primers 8F (5′-TGGAGAGTTTGATCCTGGCTCAG-3′) and 533R (5′-TACCGCGGCTGCTGCTGGCAC-3′) (Sigma Aldrich, Darmstadt, Germany) [24]. PCR reactions were performed in a volume of 50 μL, consisting of 1 U of recombinant Taq DNA polymerase (Thermo Fisher Scientific, Waltham, MA, USA), 0.2 mM each of dNTPs (Thermo Fisher Scientific, Waltham, MA, USA), 2 mM MgCl_2_, 25 pmol of each primer, and 50 ng of DNA. PCR conditions were 10 min at 94 °C followed by 30 cycles at 94 °C for 30 s, 60 °C for 30 s, and 72 °C for 45 s, and a final extension at 72 °C for 10 min. The amplification products were purified with an ExoSAP kit (Thermo Fisher Scientific, Waltham, MA, USA) and then sequenced with SeqStudio Genetic Analyzer (Applied Biosystems, Bedford, MA, USA) using the BigDye Terminator v 1.1 cycle sequencing kit (Thermo Fisher Scientific, Waltham, MA, USA). Subsequently, sequences were aligned in the GenBank database using the web-based basic local alignment tool BLAST (http://blast.ncbi.nlm.nih.gov/Blast.cgi/, accessed on 4 March 2021) to identify species with highest similarity and they were aligned against each other using a Clustal W program [25].

### 2.5. Characterization by Means of MLST Analysis

Multi-locus sequence typing analysis (MLST) was performed to assess genetic similarity between isolates [26] by analyzing the genetic profile of seven housekeeping genes, according to the pubMLST scheme (https://www.pubmlst.org/, accessed on 4 March 2021). The genes examined were: *glpF*, *ilvD*, *pta*, *purH*, *pycA*, *tpiA*, and *gmk*/*rpoD*, depending on identified species.

PCR amplifications were performed using Taq DNA polymerase recombinant (Thermo Fisher Scientific, Waltham, MA, USA), in a volume of 50 μL with the following cycling profile: a denaturation step at 94 °C for 10 min, followed by 35 cycles at 94 °C for 1 min, 54–60 °C for 1 min and 72 °C for 45 s, and final extension at 72 °C for 10 min. PCR products were purified using an ExoSAP Kit protocol (Thermo Fisher Scientific, Waltham, MA, USA) and sequenced with BigDye Terminator 1.1 (Thermo Fisher Scientific, Waltham, MA, USA). The samples were analyzed by capillary electrophoresis using SeqStudio genetic analyzer (Applied Biosystems, Bedford, MA, USA). DNA sequences obtained after amplifications were aligned and compared to sequences available on the pubMLST database (https://www.pubmlst.org/, accessed on 4 March 2021). Finally, an allele number was assigned to each gene and the combination of the seven alleles defined the sequence type (ST), which is used to categorize the target strains.

### 2.6. Analysis of Toxin Genes

Isolated colonies were tested for the presence of enterotoxin genes. Five different enterotoxins (HBL, NHE, CytK, BceT, and entFM) and one emetic toxin were examined [27]. The set of primers used, and the amplification conditions were performed, as previously described [28].

DSM4282 and DSM4312 were used as reference strains for diarrhoeagenic and emetic gene sets. The PCR products were analyzed in 2% agarose gel. The oligonucleotides were validated in simplex, and all PCR experiments were performed twice for each strain.

### 2.7. Screening of Biofilm Forming Strains Using Micro-Method Assay

Eight isolates were tested for biofilm forming ability and were selected as representative of: (i) species identified; (ii) collection’s season; and (iii) genetic profile. The experiments were performed according to the protocol previously described by Stepanovic et al. [29].

Briefly, an overnight bacterial culture grown in brain heart infusion (BHI) (Oxoid, Basingstoke, UK) was standardized to 0.5 MacFarland standard (cell concentration 10^8^ CFU/mL). Subsequently, 200 μL of inocula accurately diluted were added to 96-well polystyrene microplates (Sarstedt, Nümbrecht, Germany), in triplicate for each isolate, while the negative control wells contained the uninoculated broth. The microplates were incubated at 30 °C for 24 h in static conditions. After the incubation, the medium was discarded and each well was carefully washed three times with 300 μL of PBS sterile phosphate buffer saline solution (PBS, pH 7.3, Oxoid, Basingstoke UK) to remove unattached cells. Biofilms were heat-fixed at 60 °C for 1 h and then stained with 150 μL of 2% *w*/*v* crystal violet solution (Chem-lab, Zedelgem, Belgium) for 15 min. Unbound crystal violet was discarded by washing each well 3 times with distilled water and dried at 37 °C for 15 min. To quantify the biofilm formation, 150 μL of 95% ethanol solution (Honeywell, Charlotte, NC, USA) were added to each well, and the absorbance was measured at 595 nm on a microplate reader (iMark plate reader, Sydney, NSW, Australia). A mean of OD values was calculated for each strain (ODs) while the cut-off OD (ODc) was calculated as three standard deviations above the mean OD of the negative control [21]. The strains were classified as weak (ODc < ODs ≤ 2 × ODc), moderate (2 × ODc < ODs ≤ 4 × ODc), strong (4 × ODc < ODs), or no (ODs ≤ ODc) biofilm producers.

### 2.8. Statistical Analysis

The study of temporal trend of microbial counts was evaluated by the Wilcoxon test to compare the total bacterial median values of 16 batches during two seasons (*n* = 32). For biofilm forming ability, data were analyzed performing a one-way analysis of variance (ANOVA) followed by a Tukey’s multiple comparison test. Statistical analyses and graphing were conducted with GraphPad Prism version 8.4.3 (GraphPad Software, San Diego, CA, USA). Differences were considered statistically significant when *p* values were less than 0.05.

## 3. Results

### 3.1. Enumeration of Vegetative Bacterial Contamination and Identification of Isolates

Only viable microbiota was quantified and studied.

No vegetative anaerobic bacteria were identified during the entire considered period (a number < 10 CFU/mL was detected). Cultures in PCA showed total bacterial load of vegetative cells varied over time, with a highest density observed in summer (range of 2.0–2.4 Log CFU/mL), whereas lowest levels occurred in autumn (range of 1.3–1.6 Log CFU/mL) with statical significance differences assessed during the two seasons (*p* < 0.001) (Figure 2).

During the entire collection period, a total of 103 colonies were identified by MALDI-TOF with high confidence score. Results showed the presence of microorganisms of *Bacillus* genus, in particular *B. subtilis* and *B. cereus* (Table 1). A prevalence of *B. subtilis* was evident in all 4 weeks of autumn season and in 2/4 weeks of summer period, while during week I (summer), the number of *B. cereus* was higher compared to *B. subtilis* and, in the week III (summer), a balance of both species was highlighted (Table 1).

Moreover, to support MALDI identification, an independent molecular analysis was performed. The sequencing of 16S rRNA was assessed on thirty-five isolates, including both species identified and also representative of the collection weeks (Appendix A). Results showed a sequence’s homology >99.9% both for *B. subtilis* and for *B. cereus*, which were in accordance with MALDI-TOF MS identification.

As a result, all thirty-five isolates, fourteen *B. cereus*, and twenty-one *B. subtilis* were subjected to further characterizations (Appendix A).

### 3.2. Molecular Characterization by MLST Analysis

Basing on sequence alignment of multiple housekeeping genes, the fourteen strains of *Bacillus cereus* grouped in two different sequence types (Table 2A). The most common ST group was ST-371, which was detected in thirteen isolates, while in one isolate (BC_2), ST-32 was identified (Table 2A).

For *Bacillus subtilis*, six of seven allelic profiles matched with the existing types. A new allelic profile was identified in the *pycA* gene, and it was deposited and registered as “108” (https://pubmlst.org/bigsdb?db=pubmlst_bsubtilis_seqdef&page=alleleInfo&locus=pycA&allele_id=108, accessed on 4 March 2021). This new allele showed four SNPs of difference from the more similar allele “107”. Moreover, the presence of the new allele, and the combination of the other ones, produced a new sequence type, which was submitted and registered as “ST-249” (https://pubmlst.org/bigsdb?page=profileInfo&db=pubmlst_bsubtilis_seqdef&scheme_id=1&profile_id=249, accessed on 11 March 2021). All isolated strains of *B. subtilis* showed the same sequence type (Table 2B).

### 3.3. Prevalence of Toxigenic Determinants in Bacillus cereus Isolates

The distributions of the diarrheal enterotoxin genes and emetic toxin cerulide genes in the fourteen *Bacillus cereus* isolates are shown in Table 3.

In the present analysis, the presence of hemolysin *hbl* was not detected. All isolates were positive for all three genes coding for non-hemolysin enterotoxin *nhe*. Positivity for *entFM* and *cytK* was also detected in all analyzed bacterial isolates. In contrast, the *bceT* gene and emetic strains were not identified (Table 3).

### 3.4. Biofilm Forming Potential

Eight isolates were tested for biofilm formation, according to the criteria, as previously reported (see Section 2.7). For *B. cereus*, the strain tested and isolated with ST-32 was collected in summer (BC_2). Among another three sets of strains with ST-371, one derived from summer (BC_14) and the other two were from autumn (BC_36 and BC_44) (Figure 3). For *B. subtilis*, two isolates collected during summer (BS_8 and BS_23) and two in autumn (BS_42 and BS_54), all with the same sequence type (ST-249) were included (Figure 3).

Results showed that all isolates produced biofilms: five (5/8) were classified as strong producers (BC_2, BC_14, BS_23, BS_42, and BS_54), one moderate (BC_44), and two (2/8) weak (BC_36 and BS_8) biofilm producers, with statistical significance difference observed (Figure 3). 

## 4. Discussion

Although many scientific reports are focused on dairy products, few studies have attempted to characterize processed cheeses. An understanding of the microbiological quality of processed cheese is essential for answering questions related to the food quality and safety of these dairy products. Thus, the main objective of our study was to describe the residual microbiological population at the end of processing of processed cheeses in a dairy plant located in northern Italy.

Isolated bacterial colonies grown in PCA were identified by MALDI-TOF MS. In recent years, the use of mass spectrometry for microbial identification rapidly increased. Conventional culture methods, although remaining fundamental, are often time-consuming since the preparation of one or more subcultures for a species level identification could be necessary. In this contest, MALDI-TOF represents a suitable method for the identification of culturable microorganisms through the analysis of their protein profiles [30], with the advantage to be a fast and cost-effective technique. Moreover, microbial identification was also confirmed with an independent molecular method based on the sequence of 16s rRNA regions, with a complete concordance among the two techniques. Our results showed a homogenous bacterial population belonging to *Bacillus* genus, as both *B. subtilis* and *B. cereus* species were identified.

The presence of spore-forming bacteria, in particular *Bacillus* spp., has been previously reported in processed cheeses [3]. Several studies demonstrated the ability of spore-forming bacteria to survive after severe heat treatment [31,32,33].

In a study focused on dairy processing and food products, authors founded *B. cereus* as the most frequent species identified, followed by *B. subtilis* and *B. licheniformis* [34]. An incidence of *B. cereus* around 4.0–5.0 log CFU/mL was also reported in pasteurized and sterilized milk [16,35]. A comparable degree of contamination was highlighted in artisanal Mexican cheeses, in which the incidence of *B. cereus* was about 30% [36]. A greater prevalence (35–46%) was reported in raw milk and Ghanaian traditional milk products, but with a minor total count (2.0-3.2 log CFU/mL), while the research of *B. cereus* prevalence in soil samples was considerably higher, i.e., around 70% [33]. In an Italian dairy plant, the presence of *B. cereus* was monitored on different areas of processing and in the final product [37]. The microorganism was founded in environmental samples, while no detection was assessed in dairy product [37]. In a study by Liu et al. [38], liquid milk and milk powder showed a high prevalence of *B. cereus* around 44% and 26%, respectively. Lower percentages of *B. cereus* were founded in raw milk (26%), pasteurized milk (12%), cheese (10%), UHT milk (8%), and powdered infant formula (7%) [31].

In our study, only two *Bacillus* species were identified. This evidence could be related to the technology of the production process. Anyhow, despite species-specific differences in inactivation temperatures observed among *Bacillus* genus, i.e., in the case of *B. cereus* and *B. subtilis*, heat resistance temperatures were similar [39].

This extraordinary spore resistance could generate problems for the food industry, as the spores survive many of the currently applied food preservation treatments and may cause spoilage [3,40]. It has been reported that *Bacillus* spp., especially *B. cereus*, produced various extracellular enzymes which can alter the organoleptic quality of milk and dairy products [41]. Further studies will be necessary to assess the ability of our isolates to produce proteases and lipases.

In consideration of high diffusion of *Bacillus* spp. in the environment and in food, various molecular typing methods were used to evaluate the genetic relationship between isolates. Among them, a random amplified polymorphic DNA polymerase chain reaction (RAPD_PCR) was proposed as a tool to explore the genetic diversity in *B. subtilis* [42] and *B. cereus* [43], using short random primers to amplify multiple DNA fragments. This kind of method requires time-consuming and laborious preparation. It may also be subjected to great variability and, therefore, might be difficult to perform for large-scale analyses. Other genotyping methods are based on sequencing of specific housekeeping genes, such as *panC* [44] and *rpoB* [45], to distinguish very closely related ecological populations.

In this study, to explore the phylogenetic relationships between isolates, we used MLST analysis which is based on direct nucleotide sequencing of seven housekeeping genes with the advantages of being definitive and standardizable [26]. Results showed a high genetic similarity among the isolates. In the case of the fourteen *Bacillus cereus* included in the analysis, thirteen of them showed the sequence type 371, while in only one strain (BC_2), the ST-32 was detected. The ST-371 has been previously isolated in microorganisms derived from eggs [46] and, to our knowledge, there are no other identifications in dairy products. On the contrary, ST-32 has been described as a contaminant in milk [47] and other sources of an unspecified origin (https://www.pubmlst.org/, accessed on 4 March 2021). All twenty-one *B. subtilis* isolates showed a common sequence-type, ST-249, which is reported for the first time in this study.

This genetic homogeneity was also assessed by examining the presence of toxin genes for *B. cereus*. All the fourteen isolates showed the same toxigenic profile. Previous studies reported that nearly all strains of *Bacillus cereus* possess the genes of at least one of the diarrheal enterotoxins [27,48,49]. In our analysis, all isolates were positive for the three genes coding for non-hemolysin enterotoxin *nhe*, several authors reported that *B. cereus* strains isolated from different food sources possess the *nhe* genes [50,51,52]. Positivity for *entFM* and *cytK* was also detected in all analyzed isolates. An analogous result for the *cytK* gene was reported by Tirloni et al. [37] while a high, but not total, prevalence of both genes was described by Owusu-Kwarteng et al. [33]. In contrast, the *hbl*, *bceT* gene and emetic strains were not identified, and the results were in accordance with recent studies which underline the low frequency of emetic *B. cereus* strains in food products [33,53]. Anyhow, the assessment of genetic positivity of toxin genes in these *B. cereus* isolates is not necessarily correlate to toxins production, and, therefore, it cannot be associated to a concrete health risk [54].

The strong genetic similarity highlighted in the isolates could be related to a contamination of raw materials, to other external sources, or to the presence of a bacterial population in the processing lines.

Moreover, a role in microbial persistence could be linked to the presence of high percentages of fat in processed cheeses, which could be represented as a protective factor. In a study which monitors the manufacture and aging of cheddar cheese, characterized by different fat percentage, authors observed that the microbial load decreased faster in the group with a reduced percentage of fat than in the control group in which the fat content was higher [55]. A recent study carried out in peanut oil demonstrated that this matrix with high fat content protected bacteria from thermal inactivation by desiccation of bacterial cells with a decrease in water activity [56]. In this sense, fat may provide a microenvironment which may protect bacteria from antimicrobials.

Another hypothesis of recurring contamination could be related to the ability of *Bacillus* spp. microorganisms to form biofilm [21]. As support, we showed that tested isolates (*n* = 8) form biofilm: 5/8 classified as strong, 1/8 moderate and 2/8 weak producers. It well known that biofilms could become a reservoir of bacteria much more difficult to reach by common cleaning in place practices. Most of these cleaning systems are designed and optimized to remove milk residues, with variable efficiency in eliminating surface-adherent bacteria [57]. This limitation may lead to the accumulation of microorganisms on the equipment surfaces, resulting in biofilm formation [58], therefore constituting a source of continuous recontamination of the products.

## 5. Conclusions

In this study, a recurrent contamination of microorganism belonged to *Bacillus* spp., specifically *B. cereus* and *B. subtilis*, was highlighted in a dairy plant. Molecular characterization by MLST showed a high genetic similarity between isolates of the same species. Moreover, a study of toxigenic genes showed the same genetic profile in all *B. cereus* isolates. These results could be related to the presence of a microbial population inside the dairy plant. This hypothesis may be reinforced by the study of biofilm-forming ability, which showed that most of the isolates are characterized by a robust biofilm formation. Biofilms can be formed in different sites of processing lines, contributing to a recurrent contamination of the product. However, since it cannot be excluded, an initial high bacterial count in raw materials, further studies will be necessary to examine this hypothesis, jointly with the sampling of environmental superficies. In conclusion, the presence of *Bacillus* spp. in the final product, even if in low counts, represent a warming signal which could compromise the suitability of processed cheeses during the shelf life.

## Figures and Tables

**Figure 1 foods-10-02876-f001:**
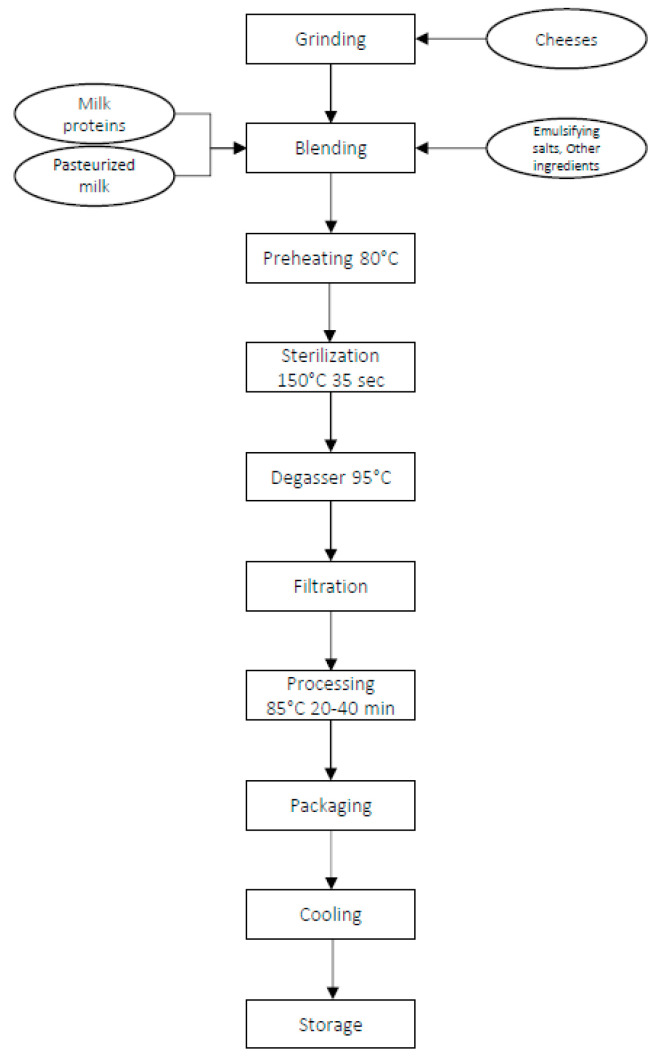
Flow chart of processed cheese production.

**Figure 2 foods-10-02876-f002:**
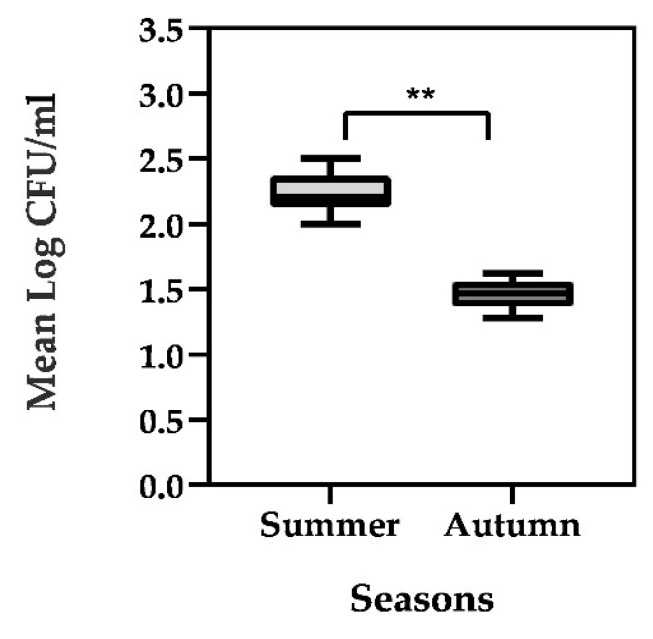
Bacterial load mean (Log CFU/mL) of vegetative cells isolated in processed cheeses during summer and autumn season. Each boxplot represents the result of a mean of sixteen batches, *n* = 16. Statistical analyses are based on the Wilcoxon test. Asterisks indicate significant differences among groups, **: *p* < 0.001.

**Figure 3 foods-10-02876-f003:**
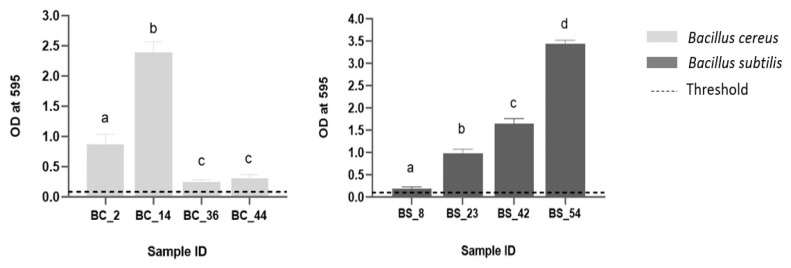
Biofilm formation of four selected *B. cereus* on the left, and four *B. subtilis* isolates on the right side. The OD threshold is reported as dotted line. Error bars indicate standard deviation. Different letters on the top of the bars indicated significant difference among the isolates, *p* < 0.05.

**Table 1 foods-10-02876-t001:** MALDI-TOF identification of bacterial colonies grown in PCA. Results showed the number of *B. subtilis* and *B. cereus* isolates and the percentage of each week.

Weeks	Season	Total Number of Colonies Identified	*Bacillus subtilis* Identified	*Bacillus cereus* Identified
I	Summer	13	5 (38.5%)	8 (61.5%)
II		12	10 (83.3%)	2 (16.7%)
III		12	6 (50.0%)	6 (50.0%)
IV		14	13 (92.9%)	1 (7.1%)
V	Autumn	19	15 (78.9%)	4 (21.1%)
VI		12	8 (66.7%)	4 (33.3%)
VII		12	10 (83.3%)	2 (16.7%)
VIII		9	5 (55.6%)	4 (44.4%)

**Table 2 foods-10-02876-t002:** MLST allelic profiles of isolated strains. (**A**) For *Bacillus cereus* 1 isolate (BC_2) showed ST-32, in the remaining 13 isolates the sequence type identified was ST-371. (**B**) All twenty-one *Bacillus subtilis* exhibited the same new sequence type (ST-249).

(A)								
Sequence Type	Allelic Type							Number of Isolates
	*glpF*	*gmk*	*ilvD*	*pta*	*purH*	*pycA*	*tpiA*	
ST-32	5	4	3	4	15	6	16	1/14
ST-371	21	2	19	5	32	18	2	13/14
(**B**)								
**Sequence** **Type**	**Allelic Type**							**Number of Isolates**
	** *glpF* **	** *ilvD* **	** *pta* **	** *purH* **	** *pycA* **	** *rpoD* **	** *tpiA* **	
ST-249	3	89	35	4	108	3	4	21/21
(new)	(new)							

**Table 3 foods-10-02876-t003:** Toxin profile of *Bacillus cereus* isolates included in the study. Results +/− indicate presence or absence of genes examined.

Strains	HBL Complex			NHE Complex			Other Enterotoxins			Emetic Strains
	*hblA*	*hblC*	*hblD*	*nheA*	*nheB*	*nheC*	*entFM*	*bceT*	*cytK*	*ces*
BC_2	−	−	−	+	+	+	+	−	+	−
BC_3	−	−	−	+	+	+	+	−	+	−
BC_5	−	−	−	+	+	+	+	−	+	−
BC_14	−	−	−	+	+	+	+	−	+	−
BC_19	−	−	−	+	+	+	+	−	+	−
BC_22	−	−	−	+	+	+	+	−	+	−
BC_24	−	−	−	+	+	+	+	−	+	−
BC_26	−	−	−	+	+	+	+	−	+	−
BC_29	−	−	−	+	+	+	+	−	+	−
BC_35	−	−	−	+	+	+	+	−	+	−
BC_36	−	−	−	+	+	+	+	−	+	−
BC_38	−	−	−	+	+	+	+	−	+	−
BC_40	−	−	−	+	+	+	+	−	+	−
BC_44	−	−	−	+	+	+	+	−	+	−

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
