# Peer review of "Characterization of Vegetative Bacillus cereus and Bacillus subtilis Strains Isolated from Processed Cheese Products in an Italian Dairy Plant"

_foods, 2021, doi:10.3390/foods10112876_

Round 1
Reviewer 1 Report
The manuscript is interesting and well written. Bacillus sporulated bacteria are especially important in foods that have undergone heating, such as processed cheeses. Some data are missing and there is a certain conceptual deficit regarding the evaluation of the microbiome of the products studied, which leads to conclusions that are not entirely correct. The observations are detailed:
1) When it is necessary to determine the viable microflora in which it is expected to find sporigenic bacteria, the presence of viable cells and spores must be considered. Both cell forms can come from direct contamination "in house" as well as from spores present in raw materials, which survive heat treatment and then germinate. For the latter case, no one can assure that the resistant spores germinate immediately and will be present as vegetative cells. As microbiologists know, all characteristics are strain dependent.
2) Therefore, faced with the inability to know if there are vegetative cells, non-germinated spores or both in the product, a microbiological analysis must include both counts: one direct in PCA, as was done, and another, also in PCA but with a previous treatment of the sample 10-15 min at 80 ° C to germinate the spores and form colonies. In the study, only a vegetative cell count is made, and from this colonies are isolated, identified and characterized. This is correct but should be clarified in the text (and perhaps in the title) that only vegetative cells of Bacillus are counted and studied
3) According to the results and discussion, it would seem that the authors only consider contamination "in house" and do not consider the possible presence of spores in the raw materials and inputs used in the process. This should be included in the text
4) Since when talking about spores in raw materials that must resist heat treatment, it is essential to clarify which heating is used in the processes of cheese plants. The meaning of "the first hours of production ..." is not well understood. Please, clarify better
5) The methodology used in points 2.3, 2.4, 2.5, 2.6, 2.7 and 2.8 is adequate
6) Results: it should be clarified that the viable microbiota was quantified and studied. Since it is difficult to go back in the study to include counts of possible spores present, it should at least be indicated what exactly was done. The viable cell counts were those that were (low in numbers) but no one knows if there were also ungerminated spores in the samples (and that do not give colonies, obviously)
7) page 9, third paragraph, line 3: "research"
Reviewer 2 Report
This manuscript covers an interesting topic related to dairy spoilage that is the characterization of Bacillus cereus and B. subtilis, using an interesting approach. Its novelty is not deep but brings important contributions to the understanding of this topic. The most important critic about this manuscript is that this study was performed in a single dairy plant and to include more industries must have improved its quality. In this condition, it should had been prepared as a communication style. Furthermore, sequencing the panC to establish the isolates phylogenetic groups would be interesting to be verified. Authors should describe the flowchart of the processed cheese sampled in the specific industry (time x temperature treatments), to allow understanding the factors that could have influenced on the high-isolation of the same STs. If authors had included other dairy plants or products, probably others ST could have been detected. Other suggestion is to test the isolates proteolytic and lipolytic capacities using specific agar plates which could improved it to be publish in this special issue related to dairy spoilage. M&M section must be more detailed as pointed below. However, I have major and minor remarks to be addressed, which will be pointed below.
Title
Authors use Bacillus but they just isolated B. cereus and B. subtilis. Why authors did not include the others species from B. cereus group?
Abstract
L10: widely used? Make it more clear
L26-28: why authors stated “resistant to normal CIP”? This hypothesis was not tested. Could not be a contamination after thermal processing from environment? Authors just sampled the cheeses and not previous steps to understand the contamination sources.
Keywords: remove dairy processing
Introduction
Well written but more detailed information about Bacillus spoilage must be included. Which enzymes are produced? Which Bacillus are able to produce them? Authors focused on Bacillus but must make clearer the importance of specific species and mechanisms. Remember to be a large group of species and authors just included two in this study.
M&M
2.1. Why authors collected the first 50 unities instead to make a better sampling strategy covering the whole production? How many samples were evaluated in total?
Please, include the flowchat used in the dairy plant to produce the processed cheeses sampled.
If authors aimed to isolate Bacillus, why they used the combination of PCA + Iron Sulfit instead of B. cereus Agar or MYP??? Please clarify the reason.
There is a lack of information about colonies selection for DNA extraction of MALDI. How were they selected? How many colonies per plate? What were the used criteria for selection? Were they passed to BHI broth (or similar) before DNA extraction?
Please, include the seven housekeeping genes used for establishing MLST
Results
Authors got 103 colonies of Bacillus but how many were tested? Include more information about it.
Table 1 – adjust the second column (hbl genes)
Authors must include information in M&M that they did not evaluate the biofilm capacity for all isolates. Why only eight were tested? Clarify the reason. Authors emphasize that the capacity was different in the same genetic patterns but just tested few isolates. The information from all others isolates must have been included.
Table 4 must be in M&M section and not results.
Discussion
The discussion section focused on extensive present the advantage of using MALDI but several information about B. cereus contamination and consequences are lacking. Authors must discuss their findings with other papers focused on B. cereus on dairy.
https://doi.org/10.3390/toxins12070454, https://doi.org/10.3168/jds.2019-17541, https://doi.org/10.1016/j.foodcont.2016.04.012, DOI 10.1186/s12866-017-0975-9 , https://doi.org/10.1089/fpd.2019.2673, https://doi.org/10.3390/microorganisms8060949
L263-265 – repetition of results and no discussion
Authors must discuss reasons for detecting only B. cereus and B. subtilis. What can explain the detection of these two species and not detecting the others from B. cereus group?? Are there differences regarding their thermal resistance?
L301 – 302 – Why? Maybe the thermal treatment? You can just suggest this but this was not proven. Authors must have evaluated other sites such as equipment, raw milk…
L317 – 320: discuss them
L320 – 328: discuss using Bacillus. To compare with other microorganisms does not make sense
Conclusion
The conclusion is not supported by the results.
Round 2
Reviewer 1 Report
I accept the responses of the authors. Only a few minor fixes remain to be made. Detail:
- Title: Modify like this: ".... vegetative Bacillus cereus and Bacillus subtilis strains isolated from ...."
- - Abstract, lines 16,17,19,21 and 24: when species are mentioned, then add "strain" or "strains". Species were not isolated and studied, but strains of two species
- 2.2, Title: remove "Vegetative". If colonies are counted, they can form only from vegetative cells (spores do not form colonies unless germinated first)
Author Response
Dear Reviewer,
- we modified the title adding “… strains isolated from…”.
- In the abstract, in all the lines when we mentioned the species, we added “strains”.
- In the section 2.2 we removed "vegetative" from the title.
Thanks for all Your suggestions.
Reviewer 2 Report
The authors performed all the suggestions and the manuscript is ready for acceptance. Congratulations.
Author Response
Thanks for all Your comments and suggestions.